# Substrate Optimization for PHB Production from Ricotta Cheese Exhausted Whey Using *Azohydromonas lata* DSM 1123

**DOI:** 10.3390/microorganisms13081917

**Published:** 2025-08-17

**Authors:** Angela Longo, Luca Sconosciuto, Michela Verni, Vito Emanuele Carofiglio, Domenico Centrone, Marianna Villano, Gaia Salvatori, Erica Pontonio, Marco Montemurro, Carlo Giuseppe Rizzello

**Affiliations:** 1Department of Environmental Biology, Sapienza University of Rome, 00185 Rome, Italy; angela.longo@uniroma1.it (A.L.); carlogiuseppe.rizzello@uniroma1.it (C.G.R.); 2EggPlant Srl, 00198 Rome, Italy; luca@eggplant.it (L.S.); vitoemanuele@eggplant.it (V.E.C.); domenico@eggplant.it (D.C.); 3Department of Chemistry, Sapienza University of Rome, 00185 Rome, Italy; marianna.villano@uniroma1.it (M.V.); gaia.salvatori@uniroma1.it (G.S.); 4Research Center for Applied Sciences to the Safeguard of Environment and Cultural Heritage (CIABC), Sapienza University of Rome, 00185 Rome, Italy; 5Department of Soil, Plant and Food Science, University of Bari, 70126 Bari, Italy; erica.pontonio@uniba.it; 6Institute of Sciences of Food Production (CNR-ISPA), National Research Council of Italy, 70126 Bari, Italy; marco.montemurro@cnr.it

**Keywords:** dairy waste valorization, polyhydroxybutyrate (PHB), bioplastic, *Azohydromonas lata*

## Abstract

Valorization of ricotta cheese exhausted whey (RCEW), a dairy by-product generated in large quantities worldwide, is essential to mitigate its environmental impact and unlock its economic potential. This study explores the use of RCEW as a substrate for polyhydroxyalkanoate (PHA) production by *Azohydromonas lata* DSM 1123. The substrate was characterized by low protein and fat contents and a relevant lactose concentration (3.81%, *w*/*v*). Due to *A. lata*’s inability to directly metabolize lactose, β-galactosidase supplementation was necessary. Mineral supplementation of pasteurized RCEW significantly improved both microbial biomass and PHA synthesis, achieving up to 25.94% intracellular PHA content, whereas pre-adaptation trials failed to enhance strain performance. Moderate nitrogen limitation in the substrate (C/N ratio 44) favored PHA synthesis (0.55 g/L) and 32.74% intracellular accumulation. Thermal treatments decreased initial microbial contamination, hence a balanced mixture of pasteurized–sterilized (75:25) substrate was used to modulate RCEW protein content without the inclusion of additional technological or chemical processing steps and without lactose loss or dilution. Bioreactor trials using optimized RCEW pre-treatment conditions led to a further increase in biomass (2.36 g/L) and PHA production (0.88 g/L), especially under fed-batch conditions. The extracted polymer was confirmed to be polyhydroxybutyrate (PHB), with high thermal stability and a molecular weight of 5.9 KDa.

## 1. Introduction

The large-scale production of petroleum-based plastics has raised global environmental concerns due to their durability and resistance to degradation. While valued for their mechanical and thermal properties, their disposal, via landfilling, incineration, or recycling, poses challenges, with landfills leading to long-term persistence and incineration releasing toxic pollutants [1,2]. The recycling of plastic is a time-consuming process that involves meticulous sorting, and it can alter the material’s properties, reducing its technological performance [3]. Additionally, recent studies have emphasized the environmental risks of microplastics, which can harm living organisms by affecting their gastrointestinal systems, potentially causing starvation and death [4,5,6].

The need to address the challenges associated with plastic waste disposal and to conserve limited oil resources has driven the development of alternative materials, such as biopolymers. Unlike conventional synthetic polymers, biopolymers are derived from renewable biomass and decomposed into minerals, biomass, CO_2_, H_2_O, and CH_4_ through microbial degradation, making them more environmentally friendly and sustainable than petroleum-based plastics [7]. The European bioplastics industry has recently begun implementing policies and practices to tackle these challenges. However, these efforts remain insufficient to fully meet the objectives of the Circular Plastics Alliance, the European Green Deal, and other related initiatives [8,9]. To achieve maximum efficiency, both upstream approaches, such as redesigning plastics, reducing consumption, and substituting with bio-based alternatives, and downstream strategies, including post-consumer recycling, composting, and degradation, should be implemented in tandem [10].

The replacement of synthetic plastics with bio-based, compostable, and biodegradable alternatives is a well-established and widely accepted solution to the environmental issues caused by plastic waste [11]. Although the feasibility of producing bio-based plastics was first demonstrated in the 1950s, they accounted for less than 1% of the 414 million tons of plastics produced in 2024. This limited adoption can be attributed to a number of factors, including the high production costs, the absence of large-scale production facilities and supply chains, difficulties in meeting performance requirements for various applications, and a lack of relevant standards and regulations [12]. Overall, the production of bio-based plastics is expected to reach approximately 6.3 million tons by 2027 [13].

Polyhydroxyalkanoates (PHAs) are microbially synthesized polyesters, intracellularly produced by a variety of organisms, including bacteria, fungi, yeasts, algae, plants, and cyanobacteria [14]. These biopolymers are formed as intracellular granules serving as carbon and energy reserves under specific stress conditions, such as nutrient limitations (e.g., nitrogen, phosphorus, or oxygen), pH variations, temperature fluctuations, and carbon source availability [15]. Polyhydroxyalkanoates are notable for their biodegradability and biocompatibility, as well as their favorable physical, thermal, molecular, mechanical, and end-use properties, making them a promising alternative to synthetic polymers [16]. The concept of biodegradability aligns with the tenets of a circular economy, providing a sustainable end-of-life solution for the management of post-consumer plastics and contributing to the reduction of the carbon footprint associated with plastics [17]. PHAs are currently produced by numerous microorganisms, including both pure and mixed microbial cultures, and have diverse applications in cosmetics, food packaging, medical devices, and agricultural products [18].

The high production cost of PHAs is primarily driven by the expense of acquiring or pretreating carbon substrates and the costs associated with downstream processing and biopolymer separation [19].

In recent years, there has been a shift in focus towards the utilization of renewable feedstocks and industrial or municipal waste as substrates for microbial PHA and PHB (the most common type of PHA) production to enhance cost-effectiveness and sustainability [20,21]. The utilization of by-products from the food industry or agro-industrial waste as carbon sources has been identified as a potentially viable solution, offering the dual benefit of reducing production costs and addressing the economic challenges associated with the management of waste streams [22]. Several food wastes, including molasses, olive mill waste, coffee waste, cooking oils [23], wasted bread [24], and dairy wastewater [25], have already been investigated as inexpensive substrates for growing PHA-synthesizing microorganisms. Among these, the main dairy-derived by-products—cheese whey, ricotta cheese exhausted whey (RCEW), and buttermilk—are globally available in large quantities, amounting to approximately 200 million tons of whey per year [26], with 7.0–9.5 million tons originating from Italy alone [27]. This situation gives rise to issues related to the large organic load being disposed of in the environment, with BOD (biochemical oxygen demand) and COD (chemical oxygen demand) values from 50 to 80 g/L [28,29], as well as the potential economic losses resulting from direct disposal costs and the lack of by-product valorization that still has potential for further upgrading.

Based on the above considerations, *Azohydromonas lata* DSM 1123 was cultivated on RCEW substrates, aiming to identify the environmental (mineral supplementation, nitrogen content, pre-adaptation with lactose) and process parameters (one- or two-stage cultivation) that could maximize PHA production. *Azohydromonas lata* DSM 1123 was chosen as the model microorganism due to its ability to accumulate PHB in a growth-associated manner, without requiring strictly defined accumulation phases [30,31]. This characteristic enables simplified cultivation strategies, particularly under suitable nutritional conditions. Moreover, its rapid growth, broad substrate tolerance, and resilience to non-sterile environments make it a suitable candidate for low-cost, scalable processes using minimally treated dairy waste streams. Hence, the optimized condition was also scaled up in a bioreactor and the biopolymer obtained characterized for its purity and composition as well as its thermogravimetric properties.

## 2. Materials and Methods

### 2.1. Microorganisms and Cultivation Conditions

*Azohydromonas lata* DSM 1123 (also coded as ATCC 29714), formerly known as *Alcaligenes latus*, was purchased from the DSMZ collection (Braunschweig, Germany) and cultivated in medium 830 (also known as R2A medium) [32], containing yeast extract, 0.50 g/L; proteose peptone (Difco no. 3), 0.50 g/L; casamino acids, 0.50 g/L; glucose, 0.50 g/L; starch, 0.50 g/L; Na-pyruvate, 0.30 g/L; K_2_HPO_4_, 0.30 g/L; and MgSO_4_ × 7 H_2_O (final pH 7.2). *A. lata* DSM 1123 was routinely propagated using a 10% (*v*/*v*) inoculum coming from a 24 h-refreshed culture and incubated in a 200 mL Erlenmeyer flask (containing 100 mL of culture) at 30 °C for 24 h under stirring conditions (180 rpm). The growth of *A. lata* DSM 1123 in the culture medium 830 was monitored by measuring cell density at 600 nm and by a microscopic count obtained using the Thoma cell counting chamber.

### 2.2. Dairy Wastewater Characterization

RCEW was supplied by EggPlant Srl. and derived from Apulian (Italy) dairy factories. RCEW was delivered under refrigerated conditions (4 ± 2 °C) within 24 h from production, characterized, and stored at −20 °C until use as substrate for *A. lata* DSM 1123 cultivation. The pH of the RCEW was determined by a FiveEasy Plus pH Meter (Mettler-Toledo, Columbus, OH, USA). Total nitrogen (TN) was determined by applying the standard ISO 8968-1 Kjeldahl-based method [33], using 6.38 as a conversion factor to calculate total proteins. Sugars (glucose, galactose, and lactose) were determined by HPLC analysis, using an ÄKTA purifier HPLC (GE Healthcare, Chicago, IL, USA) equipped with a Spherisorb-5-NH2 column (4.6 × 250, Waters, Sesto San Giovanni, Italy) and a refractive index detector (RI-101, Perkin Elmer, Shelton, CT, USA), using a solution of acetonitrile/water (ratio 65:35) as the mobile phase [34]. Freeze-dried samples were resuspended in 1 mL of acetonitrile (65% *v*/*v*) before analysis. Sugars were identified, and the calibration curves were obtained using commercial standards of lactose, glucose, and galactose (Sigma Aldrich, Milano, Italy). The total free amino acids (TFAA) concentration was determined using a Biochrom 30+ series Amino Acid Analyzer (Biochrom Ltd., Cambridge Science Park, UK) with a Li–cation-exchange column (0.46 cm internal diameter) by post-column derivatization with ninhydrin, as described by Verni et al. [35]. Fat and ash were determined through the international standard methods Gerber [36] and AOAC 945.4 [37], respectively.

The lactic acid bacteria cell density of the RCEW was determined using De Man, Rogosa, and Sharpe (MRS) (Oxoid, Basingstoke, Hampshire, UK) agar medium supplemented with cycloheximide (0.1 g/L). Plates were incubated under anaerobiosis (AnaeroGen and AnaeroJar, Oxoid, Basingstoke, Hampshire, UK) at 30 °C for 48 h. The cell density of yeasts was estimated on yeast peptone dextrose agar medium (YPDA) (Sigma-Merck, Darmstadt, Germany) supplemented with chloramphenicol (0.1 g/L) after incubation at 30 °C for 72 h. Total mesophilic aerobic bacteria were determined on plate count agar (PCA, Oxoid, Basingstoke, Hampshire, UK) at 30 °C for 48 h, and total *Enterobacteriaceae* were determined on violet red bile glucose agar (VRBGA, Oxoid, Basingstoke, Hampshire, UK) at 37 °C for 24 h. All the analyses were performed in triplicate.

### 2.3. Growth and PHA Accumulation Tests in RCEW

*A. lata* DSM 1123 cultivation tests were carried out at 30 °C in 200 mL batches placed in 0.5 L Pyrex glass Erlenmeyer flasks in stirring conditions (180 rpm). According to previous research, fermentation cycles last 72 h [30], and the pH was adjusted at 7.0 with 1M NaOH at t_0_ and at 3-h intervals. Cycloheximide (0.10 g/L) was added to all the liquid substrates to inhibit contaminating yeast growth.

A commercial β-galactosidase from *Kluyveromyces lactis* Lactozym Pure 6500 L (1320 U/mL) (Novozymes, Bagsvaerd, Denmark) was added to all substrates derived from RCEW at a concentration of 0.1% (*v*/*v*) [38]. The inoculum of the starter *A. lata* DSM 1123 was prepared by adding 10% (*v*/*v*) of an overnight culture. Lactose was monitored as described above. Proteins were determined by the Bradford method [39]. The carbon-to-nitrogen (C/N) ratio of the substrates was determined using a CN802 elemental analyzer (VELP Scientifica, Usmate Velate, Italy).

The experimental setup used for process optimization is described below and illustrated in Figure 1.

#### 2.3.1. Mineral Supplementation

Before the inoculation, RCEW was pasteurized at 63 °C for 30 min. At the end of the thermal treatment, the substrate was centrifuged (12,000× *g* for 15 min at 4 °C) and the supernatant (pRCEW) was used in growth tests, while the pellet (mainly consisting of denatured whey protein) was discarded. RCEW was supplemented (RCEW+) with 0.1% (*v*/*v*) trace solution (TS) and 10% mineral medium (MM), as previously proposed by Zafar et al. [30] and Sharma et al. [40], although small modifications were made to the original composition of the supplements based on preliminary trials evaluating 24-h growth of *A. lata* DSM 1123 in pasteurized RCEW in comparison to that observed in medium 830. The adjustments were justified by the high endogenous mineral content of RCEW [41], which allowed for reduced supplement concentrations compared to synthetic media. In detail, the TS had the following composition: FeSO_4_ × 7H_2_O (20 g/L); H_3_BO_4_ (0.3 g/L); CoCl_2_ × 6H_2_O (0.20 g/L); ZnSO_4_ × 7H_2_O, 0.03 g/L; MnCl_2_ × 4H_2_O, 0.03 g/L; (NH_4_)_6_Mo_7_O_24_ × 4H_2_O, 0.03 g/L; NiSO_4_ × 7H_2_O, 0.03 g/L; CuSO_4_ × 5H_2_O, 0.01 g/L. MM had the following composition: Na_2_HPO_4_, 9 g/L; KH_2_PO_4_, 4.0 g/L; (NH_4_)_2_SO_4_, 3.0 g/L; MgSO_4_ × 7H_2_O, 0.30 g/L; citric acid, 0.10 g/L; yeast extract, 0.15 g/L; CaCl_2_, 0.01 g/L.

#### 2.3.2. Pre-Adaptation

Aiming to verify the capability of *A. lata* DSM 1123 to adapt to dairy wastewaters and subsequently improve its performance in pRCEW, it was propagated in substrates containing lactose (and its hydrolysis products) following three different procedures: (i) in medium 830 supplemented with 3.8% (*w*/*v*) lactose and β-galactosidase (3.8-I) (one overnight incubation); (ii) in medium 830 supplemented with 3.8% (*w*/*v*) lactose for 3 consecutive overnight cultivation steps (3.8-III); (iii) in medium 830 supplemented with 3.8, 8.15, and 12.5% (*w*/*v*) lactose and β-galactosidase (3 consecutive overnight cultivation steps) (12.5-I).

#### 2.3.3. Thermal Treated Substrates

Besides pasteurization, RCEW was also subjected to sterilization (121 °C for 20 min, under a pressure of 15 psi over atmospheric pressure, sRCEW), followed by the protein pellet removal. The supernatants were used in growth tests in comparison to pRCEW.

#### 2.3.4. Effect of Organic Nitrogen Compound Levels

To verify the growth of the starter and the accumulation of PHA according to the organic nitrogen concentrations, substrates with different protein concentrations were obtained by mixing pRCEW (3.30 g/L protein) and sRCEW (0.10 g/L protein). In detail, substrates (pRCEW:sRCEW) were mixed in ratios of 25:75 to obtain RCEW-I (0.90 g/L protein), 50:50 to obtain RCEW-II (1.70 g/L protein), and 75:25 to obtain RCEW-III (2.50 g/L protein).

#### 2.3.5. Two-Stage Cultivation Trials

To enhance PHA accumulation under nutritional stress conditions, a two-stage process was set up, where the first stage included microorganism growth under nitrogen-abundant conditions, while in the second stage, nitrogen limitation of the substrate was achieved [42]. In particular, two different options were investigated.

The first, sRCEWamm/sRCEW, included the cultivation of *A. lata* DSM 1123 for 24 h in sRCEW supplemented with TS, MM, and ammonium sulphate (NH_4_)_2_SO_4_ (2 g/L). At the end of this first stage, the substrate (supplemented with nitrogen) was removed by centrifugation, and the cell pellet, under sterile conditions, was resuspended in the same volume of sRCEW supplemented with TS and MM (without ammonium sulphate supplementation) and incubated for a further 48 h.

A second option, pRCEW/sRCEW, included the cultivation of *A. lata* DSM 1123 for 24 h in pRCEW supplemented with TS and MM. Then, the substrate was removed by centrifugation and the cell pellet, under sterile conditions, was resuspended in the same volume of sRCEW supplemented with TS and MM and incubated for a further 48 h.

### 2.4. Bioplastic Production

Microbial biomass, expressed as cell dry matter (CDM), and the amount of purified PHA accumulated in the different experimental conditions were determined as previously described by Raho and colleagues [43], with some modification [24].

#### 2.4.1. Cell Dry Matter

The exhausted substrate was separated from the cells by centrifugation (12,000× *g* for 50 min at 4 °C). Then, to prevent cellular lysis, the cells were washed with a 0.9% (*w*/*v*) NaCl solution and centrifuged again. The cell pellet (CDM) was dried at 60 °C for at least 24 h and weighed.

#### 2.4.2. Intracellular PHA Content

PHA was extracted from CDM using a 12% (*v*/*v*) solution of sodium hypochlorite (NaOCl) and chloroform (CHCl_3_): 12.5 mL of chloroform and 12.5 mL of 12% sodium hypochlorite per gram of dried cell biomass. To achieve complete solubilization, the mixture was kept in a water bath at 30° C for 90 min and vortexed regularly. The organic phase containing PHA was separated and centrifuged again (12,800× *g* for 10 min at 4 °C) to remove impurities. The pellet obtained after evaporation of the organic solvent, corresponding to raw PHA, was recovered and weighed.

The intracellular PHA content (i.e., the ratio of PHA contained in the CDM) was calculated according to the following formula:% PHA/CDW = [PHA (mg/L) ÷ CDM (mg/L)] × 100

PHA was dissolved in CHCl_3_, and cold ethanol was added in a ratio of 1:10 *v*/*v*, aiming at the purification of the bioplastic. Purified PHA (corresponding to the precipitate) was recovered by centrifugation (12,800× *g* for 10 min at 4 °C) and characterized.

### 2.5. Scale-Up of PHA Production in Bioreactor

Biopolymer production by *A. lata* DSM 1123 was tested in a 2-L Lambda Minifor bioreactor (Lambda CZ, Brno, Czech Republic). The RCEW-III supplemented with TS and MM was used (1.2 L batches) under automatic pH and temperature control, set at 7.2 ± 0.01 and 30 ± 0.1 °C, respectively, for 72 h. Stirring was at 200 rpm, with a sterilized filtered air flow rate of 1.0 vvm (vessel volumes per minute, gas volumetric flow rate per unit volume of culture medium). No antifoam agent was added.

Two fermentation/cultivation options were tested, a batch fermentation (b/RCEW-III) and a fed-batch cultivation (fb/RCEW-III), with an initial volume of 1.0 L and the addition of 50% of the initial volume (0.5 L) of sRCEW (protein content: 12 mg/L vs. 330 mg/L) at 24 h.

### 2.6. PHA Characterization

#### 2.6.1. Purity and Composition by Gas Chromatography (GC) Analysis

The PHA concentration was quantified by GC by injecting 1 μL volume of organic liquid phase into an Agilent 8860 GC equipped with an HP-5 column (30 m × 320 μm × 0.25 μm). N_2_ was used as carrier gas at a flow rate of 10 mL/min, and the oven temperature was initially maintained at 100 °C for 1 min, then increased up to 130 °C at a rate of 12 °C/min and further increased up to 250 °C at a rate of 25 °C/min. The flame ionization detector (FID) was maintained at 300 °C. Samples for PHA analysis were prepared as described elsewhere [44].

#### 2.6.2. Thermogravimetric Analysis (TGA)

The PHA thermal stability was investigated by thermogravimetric analysis (TGA), using a Mettler TG 50 thermobalance equipped with a Mettler TC 10 A processor. Approximately 4 mg of sample was used for the analysis. All measurements were carried out under nitrogen flow from 25 °C to 500 °C at a heating rate of 10 °C/min.

#### 2.6.3. Average Molecular Weight and Polydispersity Index

The average molecular weight (Mw) and polydispersity index (PDI = Mw/Mn) of the extracted PHA were determined using a gel permeation chromatograph (GPC) equipped with a pump (JASCO PU-4180), a guard column and two columns in series (TSKgelG6000-HHR and TSKGel GMHHR-H), a column oven (JASCO CO-4060), and a refractive index detector (JASCO RI-4030), as previously reported by Salvatori et al. [45].

### 2.7. Statistical Analysis

All data from biochemical analyses were obtained in at least three biological replicates, and each replicate was analyzed twice. Data were subject to one-way ANOVA, and pair comparison of treatment means was performed by applying Tukey’s procedure at *p* < 0.05, using the IBM SPSS Statistics 26 (IBM Corporation, New York City, NY, USA) software.

Modeling aimed to describe cell growth and PHA accumulation (dependent variables) as a function of the independent variables (protein content and mineral supplementation). A software package (Statistica 12.5, StatSoft Inc., Tulsa, OK, USA) was used to fit the second-order model to the independent variables using the following equation:$$\gamma =\;\sum B_{i\;}\chi_{i\;}+\;\sum B_{ii\;}\chi_{i}^{2}+\sum B_{ij\;}\chi_{i\;}\chi_{j\;}\;$$
where γ is the dependent variable to be modeled, *B_i_*, *B_ii_*, and *B_ij_* are regression coefficients of the model, and *χ_i_* and *χ_j_* are the independent variables in coded values. This model allowed the evaluation of the effects of linear, quadratic, and interactive terms of the independent variable on the dependent variables. Three-dimensional surface plots were drawn to illustrate the main and interactive effects of the independent variables on the dependent ones.

## 3. Results

### 3.1. Dairy Wastewater and Derived Substrates Characterization

In Table 1, the main chemical and microbiological features of RCEW are reported. RCEW was characterized by a pH of 5.2, low protein and fat concentrations (0.38 and 0.20% *w*/*v*, respectively), and a considerable lactose content (3.81% *w*/*v*). Lactic acid bacteria and yeast densities were respectively 4.56 and 3.21 Log cfu/mL, while no *Enterobacteriaceae* were found.

### 3.2. Growth and PHA Accumulation Tests

#### 3.2.1. Mineral Supplementation and Thermal Treatments

*A. lata* DSM 1123 was initially cultivated in pRCEW without any supplementation except for β-galactosidase. Based on the estimated cell dry matter (CDM), the suitability of the substrate to microorganisms’ growth was confirmed (Table 2). Residual glucose and galactose were found at the end of the 72 h of incubation (0.27 ± 0.02% and 1.29 ± 0.03%, respectively). PHA accumulation corresponded to 0.13 g/L (16.19% intracellular PHA content) in pRCEW. Mineral supplementation proved significantly effective for both cell biomass, which increased by 93%, and PHA synthesis, with a production level of 0.41 g/L. In particular, the microorganism accumulated PHA amounting to 25.94% of its dry cell weight. Based on the results, MM and TS supplementation were applied to all the further cultivation tests.

A second thermal treatment option, sterilization, was applied to RCEW as an alternative to pasteurization. The primary objective of this treatment was to induce the depletion of whey proteins in the substrates, as they undergo thermal flocculation and are subsequently removed by centrifugation. Indeed, while no differences (*p* > 0.05) in lactose content were observed in the substrate, the protein content markedly (*p* < 0.05) decreased (by three times) in sRCEW compared to the corresponding pasteurized substrate (Table 2). With sterilization and protein removal, the C/N ratio markedly (*p* < 0.05) increased in both substrates (1019 vs. 34 in RCEW). The substantial removal of nitrogenous compounds in the substrates led to a significant (*p* < 0.05) reduction in growth, accompanied by a marked decrease in PHA synthesis and intracellular accumulation, with concentrations falling below 0.01 g/L (Table 2).

#### 3.2.2. Pre-Adaptation

Aiming at assessing whether the preliminary growth of the microorganism in the presence of consistent levels of glucose and galactose (resulting from lactose hydrolysis via β-galactosidase added to the culture medium) could enhance performance upon inoculation into dairy wastewaters, pre-adaptation tests were performed. The microorganism cultivated in medium 380 supplemented with 3.8% lactose for a single overnight growth cycle yielded a CDM value comparable (*p* > 0.05) to that previously observed under pRCEW+ conditions, although with a significant decrease in both PHA synthesis and intracellular content (Table 2).

The other pre-adaptation conditions tested, involving multiple cultivation cycles in medium supplemented with either constant (3.8%) or increasing (3.8%, 8.15%, and 12.5%) initial concentrations of lactose, showed a clearly opposite trend. Specifically, an increase in CDM was accompanied by a substantial reduction in intracellular PHA content (Table 2).

Based on these results, which did not demonstrate improvements over the conditions in which the inoculum was obtained using medium 830, no further pre-adaptation of the strain was employed in the subsequent cultivation tests.

#### 3.2.3. Modulation of Protein Concentration

To evaluate the effect of different protein concentrations on the growth and PHA production of *A. lata* DSM 1123, mixtures were prepared using RCEW previously subjected to either pasteurization or sterilization, resulting in reduced protein concentrations compared to the native RCEW. The reduction was 76%, 55%, and 34% for RCEW-I, RCEW-II, and RCEW-III, respectively.

None of the three tested concentrations appeared to be limiting the growth, which was similar across all conditions (*p* > 0.05). However, as the protein concentration decreased, an increase in PHA accumulation within the biomass was observed, reaching 20.96% in RCEW-II and 32.74% in RCEW-III. In the latter condition, PHA production reached 0.55 g/L (Table 2). According to the results obtained, the RCEW-III substrate was used for further experiments, namely two-stage cultivation and a scale-up in a bioreactor.

#### 3.2.4. Two-Stage Cultivation Trials

When compared to the results obtained with RCEW-III, the two-stage cultivation trials led to overall significantly (*p* < 0.05) lower CDM, particularly when the first phase was carried out in sRCEW supplemented with ammonium sulphate (−35%, Table 2). Under this condition, PHA synthesis was extremely limited. On the contrary, in the pRCEW/sRCEW trial, PHA accumulation was considerably (*p* < 0.05) higher, but it did not reach the productivity observed in the previous RCEW-III test, which was then used for further trials.

### 3.3. Scale-Up Under Bioreactor Conditions

The cultivation of *A. lata* DSM 1123 on RCEW-III was monitored in a bioreactor under constant pH and forced aeration conditions and, in one case, with a fed-batch process. Under these conditions, the CDM increased significantly, reaching up to 40% higher values compared to the in-flask condition (Table 3), whereas PHA production increased by 36% and 60% in the case of the fed-batch process. In the latter case, a significant increase (*p* < 0.05) in intracellular PHA content was also observed (+13% compared to RCEW-III and b/RCEW-III) (Table 3).

### 3.4. Characterization of the PHA

The PHA samples synthesized by *A. lata* DSM 1123 in RCEW-III were characterized. The GC analysis of the PHA extracted from the CDM corresponded to 30.6 ± 1.04% (*w*/*w*), thus confirming that RCEW promoted the PHA storage process. The analysis further demonstrated that the composition of the synthesized polymer was mainly polyhydroxybutyrate (PHB), with hydroxybutyrate (HB) content corresponding to 94.4 ± 1.20% (*w*/*w*). Then, PHB stored from *A. lata* was also characterized in terms of molecular weight, which accounted for 5920 Da and a PDI of 1.31.

The thermal stability of the PHB was investigated by TGA (Figure 2), which revealed a degradation step occurring between approximately 250 °C and 320 °C, thus showing that the PHB extracted from *A. lata* exhibits high thermal stability with a maximum decomposition rate temperature (TdMAX) of 297 °C.

This result is consistent with the known decomposition behavior of highly purified or crystalline PHB [46,47].

## 4. Discussion

*A. lata* is a Gram-negative facultative autotroph capable of using knallgas and organic molecules (including sucrose and glucose) for growth and bioplastic production, characterized by a very high volumetric productivity of PHA [31]. As previously demonstrated, *A. lata* is not able to directly use lactose as the sole carbon source [23]. This was also confirmed in this work, where different pre-adoption media were tested, and the use of lactose without galactosidase led to the lowest values of CDM and PHB.

In this work, the PHA production of the well-characterized *A. lata* strain DSM 1123 using RCEW as a substrate for its cultivation was investigated. A one-variable-at-a-time approach was used to optimize a suitable biotechnological process for PHA production in the dairy wastewaters, aiming at identifying key operational parameters affecting cell growth and polymer accumulation. This approach, although overall limited in its ability to detect interactions among variables, allows a straightforward evaluation of individual factors and is particularly useful in the development of processes intended to be as simple and cost-effective as possible to manage.

According to previous findings, *A. lata* does not benefit from very high concentrations of sugar and nitrogen in the medium [48], thus suggesting that RCEW could serve as a suitable substrate for its cultivation. Mineral supplementation significantly boosted both cell growth and PHA synthesis in pRCEW+, indicating the importance of balanced nutrient availability for optimizing microbial PHA production. This is in line with studies that emphasize the role of trace elements, particularly magnesium, potassium, and phosphate, in enhancing PHA yields [49].

*A. lata* is potentially able to accumulate PHB up to 80% of its dry cell weight [50], and it has been reported that nitrogen limitation in the media is the key parameter to promote PHB accumulation [51], increasing the efficiency of the bioprocessing and of the downstream processing.

Thermal treatments of the RCEW, which is characterized by whey proteins as the most abundant nitrogenous compounds, highlighted their crucial role in PHA production. Overall, thermal treatment, such as pasteurization, is crucial to promote the dominance of *A. lata* over contaminating microorganisms, like lactic acid bacteria and yeasts, which are present at high densities in the wastewater after ricotta cheese production. In this work, the need for thermal treatment has also been considered effective to modulate the protein content in dairy wastewater intended for use as a substrate for the cultivation of PHA-producing microorganisms, without the inclusion of additional technological or chemical processing steps, which would imply high infrastructure and operational costs [43], and without lactose loss or dilution.

Sterilization, which precipitated most of the proteins, led to significant reductions in both microbial biomass and polymer accumulation if applied to the entire volume of the cultivation substrate. These findings align with previous work showing that nitrogen limitation can trigger PHA synthesis, but extreme nitrogen depletion may instead hinder growth and reduce overall productivity [52,53]. Indeed, aiming at selecting the optimal substrate formulation, all the data resulting from the media biochemical characterization (independent variables) and cell growth and intracellular PHA accumulation (dependent variables) were factored into three-dimensional surface plots (Figure 3). Modulating the protein content of the substrate revealed that moderate nitrogen limitation is beneficial for intracellular PHA accumulation. A progressive reduction of protein concentration, obtained by adequately mixing pasteurized and sterilized RCEW, correlated with increased intracellular PHA content, reaching up to 32.74% (dark red surface in Figure 3B). This supports the well-established model in which nitrogen limitation combined with abundant carbon triggers PHA biosynthesis as a carbon and energy storage strategy [54]. Still, extreme nitrogen depletion along with low levels of mineral supplementation impair both cell dry matter and PHA accumulation (green surface in Figure 3A,B).

Several studies have investigated the effects of the substrate C/N ratio on both *A. lata* DSM 1123 growth and PHA accumulation, finding that the optimal C/N values largely depend on the substrate complexity and the cultivation conditions [23,55,56,57,58]. Wisuthiphaet and Napathorn [53] reported that a C/N ratio of 20 resulted in the highest specific growth rate of *A. lata* DSM 1123 in a sugarcane-derived substrate but significantly suppressed PHB accumulation, while progressively higher values approaching 200 were found to enhance PHB accumulation under fed-batch conditions. Under the conditions used in this study (a single 72-h batch cultivation), the optimal C/N ratio of the RCEW for PHB production was 44. Higher values (up to 117), although not affecting the overall cell biomass, did not promote PHB accumulation. These conditions clearly require a well-calibrated nitrogen supply that is adequate to sustain exponential microbial growth during the initial 24 h but becomes limiting thereafter to trigger PHB accumulation.

Pre-adaptation trials, aimed at enhancing strain performance through prior exposure to lactose or its hydrolysis products, did not lead to significant improvements in PHA production. In fact, longer or more intensive adaptation cycles even reduced PHA yield. This observation suggests that *A. lata* does not benefit from carbon source pre-conditioning under the tested conditions, differing from other PHA-producing strains where pre-adaptation can enhance metabolic efficiency [59]. Considering that the recent characterization of *A. lata* confirmed that evolutionary adaptation could be exploited to improve PHB production, it is conceivable that the adaptation conditions would need to be especially intense and prolonged in order to produce significant effects on the microorganism’s PHB-related pathways in response to the substrate [60].

Similarly, the two-stage fermentation processes investigated in this study did not result in improved PHA yields compared to optimized single-stage cultivation on RCEW. In particular, the use of sRCEW supplemented with ammonium sulphate in the first stage significantly impaired both growth and polymer synthesis. This highlights the importance of carefully balancing nitrogen availability and avoiding abrupt shifts in nutrient profiles. Although multiple parameters should be evaluated to clarify these results, it has been recently reported that *A. lata* has a peculiar response to nitrogen limitation compared to other PHA-producing bacteria [31]. It was indeed demonstrated that, unlike typical PHA-accumulating strains such as *Cupriavidus necator*, which require severe nitrogen limitation to trigger polymer accumulation in the stationary phase, *A. lata* has the ability to initiate PHB synthesis even under moderate nitrogen concentrations during active cell growth. This suggests a growth-associated pattern of PHB accumulation, potentially linked to distinct regulatory mechanisms and enzyme kinetics involved in carbon and nitrogen assimilation. As a result, *A. lata* appears less dependent on strict nitrogen depletion, which may broaden its applicability in bioprocesses using a suitable nitrogen concentration [61].

Scaling up the process in a bioreactor under controlled conditions (stable pH, aeration) led to marked improvements in both biomass and PHA production. Notably, the fed-batch trial further enhanced PHB production by 60% compared to flask cultivation, confirming the benefits of gradual substrate feeding strategies largely observed in similar microbial systems [62]. Under the optimized conditions, the conversion of lactose (initial concentration: 38 g/L) to PHB was approximately 0.02 g PHB/g lactose. Although this yield is lower than values reported in the literature (0.05 g PHB/ g lactose) [23], the simplified operational conditions adopted in this study, specifically the reduced need for pretreatments and supplementation, are intended to offset the lower yields by significantly reducing process and management costs.

The GC characterization confirmed that the accumulated polymer was PHB, characterized by a high degree of crystallinity [63]. In particular, the Mw of the extracted PHB (5.92 kDa) appears to be relatively low compared to PHB produced by other microorganisms or the same in different cultivation conditions [64,65]. Low molecular weight may limit mechanical performance, such as tensile strength and elasticity, unless the polymer is blended or chemically modified [66]. The narrow PDI value (1.31) indicates a relatively uniform polymer chain length distribution, which is favorable for reproducible material properties and controlled degradation rates.

Thermogravimetric analysis showed a maximum decomposition rate temperature of 297 °C, which is indicative of high thermal stability and consistent with the behavior of crystalline and relatively pure PHB [46,47]. Indeed, it was previously reported that PHB degradation in the range of 250–300 °C is observed for pure polymeric fractions [67].

*A. lata* is reported as one of the most efficient PHB producers in the literature, although it does not always achieve the highest yields when compared to other microorganisms. However, its ability to accumulate PHB during the exponential growth phase (whereas in other microorganisms, such as *C. necator*, accumulation occurs almost exclusively during the stationary phase after nitrogen depletion) offers the advantage of enabling simpler one-stage production processes [61,68,69]. Moreover, it is more suitable for waste-derived or complex substrates that only require mild pre-treatment, which helps lower process costs. Unlike other species, *A. lata* does not require pre-adaptation to the substrate to optimize its metabolism, contributing to greater operational simplicity in bioreactor-based processes.

Data confirmed that *A. lata* DSM 1123 is capable of producing PHB with acceptable thermal and structural characteristics under the tested conditions. Although the molecular weight is lower than optimal for certain high-performance applications, the polymer exhibits good compositional homogeneity and thermal stability.

Based on these considerations and the results obtained, the use of RCEW as a cultivation substrate for *A. lata* DSM 1123, subjected to an easy, scalable (and multipurpose) thermal pre-treatment process for modulating C/N ratios, appears to be a viable option for sustainable bioplastic production, particularly within integrated waste valorization systems.

To further improve the efficiency and, more importantly, the scalability of the process, future research should address the key operational challenges typical of non-dedicated bioplastic industrial environments, such as dairy processing plants or wastewater treatment facilities. These include high microbial contamination, fluctuations in lactose and nitrogen contents, and limited control over pH, aeration, and temperature. Investigating the impact of these factors on PHB production at an industrial scale will be essential. Additionally, a comprehensive techno-economic assessment will be necessary to evaluate the cost–benefit ratio of process implementation under such conditions.

## Figures and Tables

**Figure 1 microorganisms-13-01917-f001:**
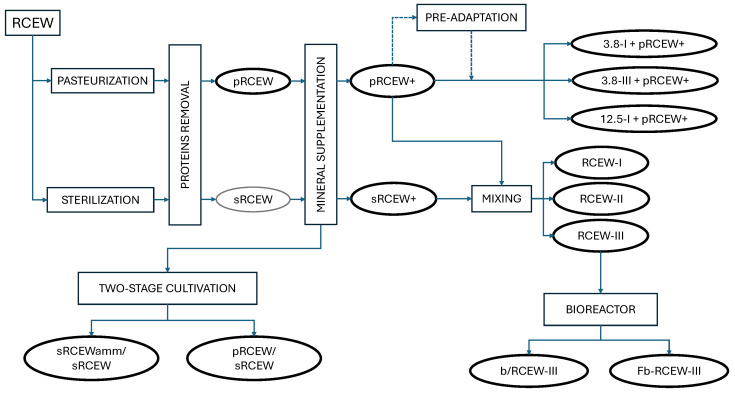
Schematic representation of the experimental setup and sample codes used in the study. The diagram outlines the processing steps of RCEW, including protein removal and mineral supplementation, and the subsequent use in cultivation trials.

**Figure 2 microorganisms-13-01917-f002:**
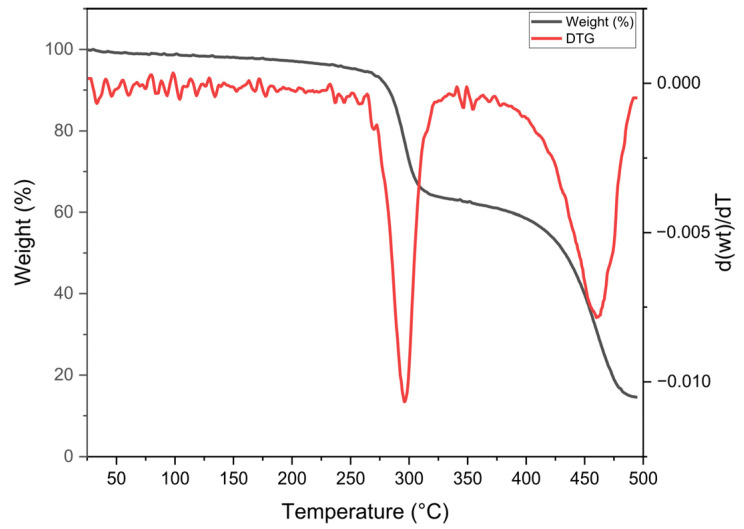
Thermogravimetric analysis curves of PHB produced by *A. lata* DSM 1123 in RCEW. In black, the weight loss (%); in red, the derivative thermogravimetric (DTG) curve, namely the weight derivative as a function of temperature (dTGA/dT).

**Figure 3 microorganisms-13-01917-f003:**
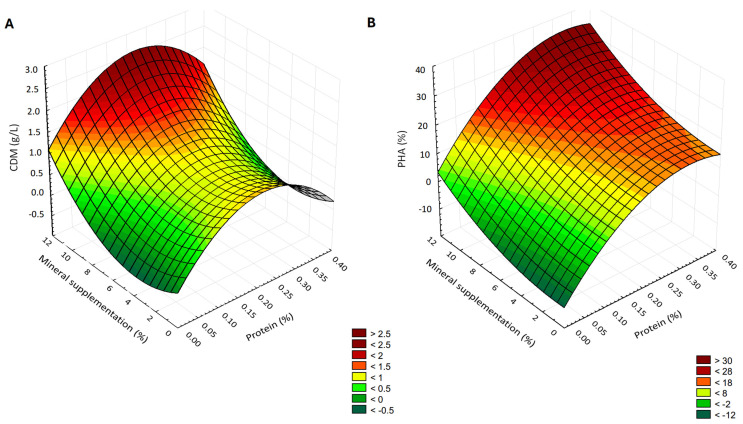
Three-dimensional surface plot of the effect of protein content and mineral supplementation on cell dry matter (**A**) and intracellular PHA content (**B**).

**Table 1 microorganisms-13-01917-t001:** Proximal composition and microbiological analysis of RCEW.

Chemical Analysis	
pH	5.2 ± 0.10
Total nitrogen % (*w*/*v*)	0.06 ± 0.01
Proteins % (*w*/*v*)	0.38 ±0.06
Total free amino acids (mg/L)	403 ± 25
Lactose % (*w*/*v*)	3.81 ± 0.15
Glucose % (*w*/*v*)	<0.01
Galactose % (*w*/*v*)	<0.01
Fat % (*w*/*v*)	0.20 ± 0.03
Ash % (*w*/*v*)	1.20 ± 0.05
**Microbiological Analysis**	
Total bacteria (Log cfu/mL)	5.30 ± 0.11
Lactic acid bacteria (Log cfu/mL)	4.56 ±0.21
Yeasts (Log cfu/mL)	3.21 ± 0.2
*Enterobacteriaceae* (Log cfu/mL)	<1

The data are the means of three independent experiments ± standard deviations (*n* = 3).

**Table 2 microorganisms-13-01917-t002:** Cell dry matter (CDM), PHA concentration, and intracellular PHA content (% PHA/CDM) obtained by *A. lata* DSM 1123 cultivated for 72 h at 30 °C in: pasteurized RCEW, sterilized and supplemented RCEW (sRCEW+); mixtures of pRCEW+:sRCEW+ in ratios of 25:75 (RCEW-I), 50:50 (RCEW-II), and 75:25 (RCEW-III) for 24 h in sRCEW+ ammonium sulphate (2 g/L) and in sRCEW+ for a further 48 h (sRCEWamm/sRCEW); cultivated for 24 h in pRCEW+ and in sRCEW for a further 48 h pRCEW/sRCEW; and in bioreactor in RCEW-III as batch (b-RCEW-III) and fed-batch cultivation (addition of 50% of the initial volume sRCEW after the first 24 h). Lactose, protein content and C/N ratio of the substrates are also reported. All the substrates were supplemented with β-galactosidase.

Adaptation	Substrate	Lactose (% *w*/*v*)	Protein (% *w*/*v*)	C/N	CDM (g/L)	PHA (g/L)	Intracellular PHA (%)
*Mineral supplementation and thermal treatments*							
-	pRCEW	3.80 ± 0.10 ^a^	0.33 ± 0.03 ^a^	34 ^e^	0.82 ± 0.15 ^c^	0.13 ± 0.05 ^c^	16.19 ± 0.12 ^c^
-	pRCEW+	3.80 ± 0.15 ^a^	0.33 ± 0.03 ^a^	34 ^e^	1.58 ± 0.12 ^a^	0.41 ± 0.03 ^b^	25.94 ± 0.54 ^b^
-	sRCEW+	3.78 ± 0.03 ^a^	0.01 ± 0.01 ^d^	1019 ^a^	0.39 ± 0.10 ^d^	0.01 ± 0.03 ^d^	2.50 ± 0.48 ^e^
*Pre-adaptation*							
3.8-I	pRCEW+	3.81 ± 0.10 ^a^	0.33 ± 0.02 ^a^	34 ^e^	1.59 ± 0.09 ^a^	0.23 ± 0.03 ^bc^	14.46 ± 0.48 ^c^
3.8-III	pRCEW+	3.79 ± 0.12 ^a^	0.33 ± 0.02 ^a^	34 ^e^	1.69 ± 0.11 ^a^	0.07 ± 0.01 ^d^	4.35 ± 0.55 ^e^
12.5-I	pRCEW+	3.80 ± 0.10 ^a^	0.33 ± 0.01 ^a^	34 ^e^	1.74 ± 0.13 ^a^	0.07 ± 0.02 ^d^	4.05 ± 0.42 ^e^
*Modulation of the protein concentration*							
-	RCEW-I	3.80 ± 0.11 ^a^	0.09 ± 0.01 ^d^	117 ^b^	1.65 ± 0.12 ^a^	0.12 ± 0.05 ^c^	7.27 ± 0.21 ^d^
-	RCEW-II	3.79 ± 0.09 ^a^	0.17 ± 0.0 ^c^	63 ^c^	1.67 ± 0.10 ^a^	0.35 ± 0.09 ^b^	20.96 ± 0.54 ^b^
-	RCEW-III	3.79 ± 0.10 ^a^	0.25 ± 0.02 ^b^	44 ^d^	1.68 ± 0.10 ^a^	0.55 ± 0.05 ^a^	32.74 ± 0.15 ^a^
*Two-stage cultivation*							
-	sRCEW_amm_/sRCEW	3.80 ± 0.10 ^a^/3.79 ± 0.11 ^a^	* 0.01 ± 0.01 ^d^/0.01 ± 0.01 ^d^	23 ^f^/1021 ^a^	1.10 ± 0.12 ^b^	0.04 ± 0.05 ^a^	3.60 ± 0.12 ^e^
-	pRCEW/sRCEW	3.78 ± 0.10 ^a^/3.79 ± 0.10 ^a^	0.33 ± 0.03 ^a^/0.01 ± 0.01 ^d^	34 ^e^/1021 ^a^	1.46 ± 0.10 ^a^	0.34 ± 0.07 ^b^	23.3 ± 0.18 ^b^

* Ammonium sulphate 2 g/L was added to sRCEW in the first stage of cultivation. The data are the means of three independent experiments ± standard deviations (*n* = 3). ^a–f^ Values in the same column with different superscript letters differ significantly (*p* < 0.05).

**Table 3 microorganisms-13-01917-t003:** Cell dry matter (CDM), PHA concentration, and intracellular PHA content (% PHA/CDM) obtained by *A. lata* DSM 1123 cultivated for 72 h at 30 °C in bioreactor in RCEW-III as batch (b-RCEW-III) and fed-batch cultivation (addition of 50% of the initial volume sRCEW after the first 24 h). Lactose, protein contents, and C/N ratios of the substrates are also reported. All the substrates were supplemented with β-galactosidase.

Substrate	Lactose (% *w*/*v*)	Protein (% *w*/*v*)	C/N	CDM (g/L)	PHA (g/L)	Intracellular PHA (%)
b/RCEW-III	3.80± 0.10 ^a^	0.25 ± 0.02 ^a^	44 ^a^	2.27 ± 0.21 ^a^	0.75 ± 0.10 ^ab^	32.83 ± 0.17 ^b^
fb/RCEW-III	3.81 ± 0.10 ^a^	0.25 ± 0.02 ^a^	44 ^a^	2.36 ± 0.20 ^a^	0.88 ± 0.11 ^a^	37.29 ± 0.21 ^a^

The data are the means of three independent experiments ± standard deviations (*n* = 3). ^a,b^ Values in the same column, with different superscript letters, differ significantly (*p* < 0.05).

## Data Availability

The original contributions presented in this study are included in the article. Further inquiries can be directed to the corresponding author.

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
