# Peer review of "Substrate Optimization for PHB Production from Ricotta Cheese Exhausted Whey Using Azohydromonas lata DSM 1123"

_microorganisms, 2025, doi:10.3390/microorganisms13081917_

Round 1

Reviewer 1 Report

Comments and Suggestions for Authors

Comments and suggestions can be found in the PDF file

Author Response

Microorganisms-3784239-peer-review-v1

Title: Optimization of Biotechnological Processes for Polyhydroxy Butyrate (PHB) Production From Ricotta Cheese Exhausted Whey Using Azohydromonas lata DSM1123

Comment: First of all, I would like to congratulate the authors on the study and manuscript submitted. The study demonstrates the efficient production of biodegradable PHB by Azohydromonas lata DSM1123 using ricotta cheese-exhausted whey (RCEW), a scarcely explored dairy by-product. It introduces a simple and scalable strategy to optimize the C/N ratio.

The authors thank the reviewer for the comments and suggestion. A point-by-point revision was provided.

Following, I provide some comments that I consider can improve the manuscript.

Major comments

  • In the title and keywords, the term PHB is used, while in the abstract the broader term PHA is consistently mentioned, except in line 31 where PHB appears. Since PHB is a specific type of PHA, this distinction should be clearly explained in the abstract and the rest of the text and figures. I recommend consistently using PHB if the focus is only on this polymer to avoid confusion for readers unfamiliar with the terminology.

As suggested, a brief explanation about the distinction between PHA and PHB was added in the introduction, nevertheless, since the confirmation that the synthesized polymer was mainly PHB was obtained only after the purity and composition analysis by gas chromatography, up until that point, in the results, the generic term PHA was used.

  • The last paragraph of the introduction is vague as neither the optimized variables, not the scale-up study nor PHB properties (or PHA, it is not clear) are mentioned.

The paragraph was clarified and the experimental plan better laid out.

  • 3.1. Mineral supplementation (Lines 171-173) – The authors mention: “Trace Solution (TS) and 10% Mineral Medium (MM), as previously proposed by 171 Zafar et al. [34], Sharma et al. [37], although small modifications were made to the original composition of the supplements”. Could the authors clarify the reason for these modifications? Was the adjustment based on bibliographic evidence, preliminary trials performed during the current study, or other considerations? Why those concentrations? Providing this rationale would strengthen the methodological transparency.

We thank the reviewer for the helpful comment. The modifications made to the original composition of the mineral supplements were based on preliminary trials conducted during the current study. These trials evaluated the 24-hour growth capacity of the microorganism in thermally treated RCEW. The results indicated that only moderate supplementation was necessary to support efficient growth, likely due to the naturally high mineral content of RCEW. Compared to synthetic media, this allowed us to reduce the concentration of added mineral supplements without compromising microbial performance. To clarify this aspect, a new comment with a proper reference was added to the revised manuscript (lines 179-182).

  • I would suggest adding a final paragraph outlining potential next steps or future research directions. As a non-expert in biopolymer production, I am particularly curious about how key operational parameters, such as aeration rate (e.g., different vvm values) or the absence of pH control, might affect PHB production, since these were briefly mentioned in the manuscript. If relevant, please consider discussing these or other factors that could be explored in future work to optimize the process further.

Thank you for the suggestion. A paragraph outlining future research perspectives has been added to the revised manuscript, with particular attention to process scale-up and the potential impact of fluctuating operational parameters under industrial conditions (lines 541-548).

Minor comments

  • Please, do not start sentences with an abbreviation like in Line 18 (RCEW) or 74 (PHAs).

Ok.                                                                                                                                                                     

2) Lines 52-53 – The text says: “biopolymers are renewable, non-toxic, and environmentally friendly”. As far as I know, not all biopolymers are necessarily renewable, non-toxic, and environmentally friendly. Please, rewrite the sentence.

The sentence was clarified as suggested and a reference added.

3) Please, combine the paragraph from Line 80 to 82 with the previous one. It could be fine to do it before the second sentence (Line 71).

As suggested, the paragraph was combined to the previous one.

4) Line 98 – The text mentions: “This situation gives rise to issues related to the large organic load being disposed of in the environment…”. Please, provide the ranges of COD and BOD.

Ok. COD and BOD values were included (lines 96-97).

5) In Line 108, the strain name is DSM1123, whereas, in Lines 113 and 115 is DSM 1123 (with a space). Please, use one format throughout the text.

The correct form, as reported on the DSMZ website, is with a space, hence the strain name was aligned within the text.

6) The same comment for the medium: DSM 830 medium (Line 110) and 830 medium (Line 115)

The name of the medium, according to DSMZ is “830”. The name was aligned within the text.

7) 2.1. Microorganisms and cultivation conditions. Lines 113-115: “A. lata DSM 1123 was propagated daily using a 10% (v/v) inoculum coming from a 113 24 h-refreshed culture and incubated in Erlenmeyer flask at 30 °C for 24 h under stirring 114 conditions (180 rpm)” – In my opinion, the sentence is a bit ambiguous since “…was propagated daily…” is not clear for me. Did you take 10% (v/v) of inoculum from a 24h fermentation to inoculate another DSM 830 medium? Did you grow the strain in a preculture medium for 24h and transfer it to perform the fermentation? Or, all of this process it is to maintain the strain? On the other hand, please provide the working and flask volume.

The daily propagation was necessary to maintain the strain alive and consisted in inoculating 10% of the 24h-fresh culture in a new sterile medium. The sentence was clarified.

8) 2.2. Dairy wastewater characterization (Line 121) – I think that the reference 28 is not correctly placed. Please, check it

The reviewer is correct; the reference was unnecessary.

9) 2.2. Dairy wastewater characterization (Line 140) – It seems that there is an extra space: “incubated, Xunder anaerobiosis”.

Done.

10) The same comment in Line 156: “…adding 10X% (v/v) of…”. Please, check the rest of the text.

The sentence is correct. Whereas in the previous paragraph, the daily propagation to keep the strain alive was described, this paragraph describes the procedure to inoculate the strain in RCEW. Indeed, the procedure was identical to the routine propagation except for the fact that the 10% inoculum was not added in sterile 830 medium but in RCEW.

11) 2.3.4. Two-stage cultivation trials (Line 211) – Sorry, in the sentence “… the substrate was removed by centrifugation”, I do not understand what the substrate removed is. Do you mean the medium by centrifugating and separating liquid and solid? Please, clarify it

Yes, the substrate - exhausted RCEW supplemented to provide nitrogen-abundant conditions in the first step - was removed by centrifugation and then replaced with RCEW providing low nitrogen conditions. The sentence was clarified in the text.

12) 2.7. Statistical analysis (Lines 278-279) – Could you provide which are the independent variables, please?

Independent variables were protein content and mineral supplementation. This aspect was included in the text.

13) Please ensure consistent formatting of units throughout the manuscript. For example, in Line 254, the heating rate is reported as “25(°C/min”, whereas in Line 261 it appears as “10(°C(min-1”. A uniform style should be adopted for clarity and standardization.

Ok. The units were standardized throughout the text.

14) Please, consider to add a table to summarize all the trials performed. In that table, readers can find the different stages of the study with the culture media used (supplemented or not) and operating conditions. Moreover, you could add which are the independent variables studied as well. In this case, Comment 12 can be omitted.

According to the suggestions (also comment 12) new comments have been added to the revised text to better identify the independent variables.

15) 3.2.1. Mineral supplementation and thermal treatments. The reported PHA concentrations were 0.13 g/L and 0.46 g/L, presumably obtained from an initial lactose concentration of 38.1 g/L. However, the theoretical yield of PHA from lactose (expressed in g/g or mol/mol) is not discussed. Including this value at least once would allow readers to assess the efficiency of the conversion. For example, a yield of 0.46 g/L from 38.1 g/L corresponds to approximately 0.012 g PHA/g lactose, which seems quite low. Providing the theoretical or maximum expected yield would help contextualize whether this result is satisfactory.

Thank you for the observation. As also reported in the literature, the lactose-to-PHA conversion yields obtained with Azohydromonas lata DSM 1123 are generally higher than those achieved in this study, although not drastically so (e.g., approximately 0.05 g PHA/g lactose vs. ~0.02 g PHA/g lactose obtained in our bioreactor experiments). However, our primary objective was to minimize the need for extensive pretreatments and supplementation, aiming for a simplified and more cost-effective process. While the conditions of this studiy resulted in lower conversion yields, it supports the broader goal of reducing overall process complexity and cost. Further studies can focus on maximizing the yields.

New comments with a proper reference have been added to the revised text (lines 504-509).

16) 3.2.1. Mineral supplementation and thermal treatments (Line 307) – Please, specify that “no difference (p<0.05) in lactose” is lactose content

Ok.

17) 3.2.2. Pre-adaptation (Line 320) – “…cultivated in medium supplemented with 3.8% lactose…”. Please, specify the medium: DSM 380, sRCEW, pRCEW… The same in Line 325

As reported in the material and methods section, the pre-adaptation with lactose supplementation was performed on medium 380. This aspect was specified in the results as well.

18) 3.2.3. Modulation of protein concentration (Lines 340-341) – “an increase in PHA accumulation within the biomass was observed, reaching 20.96% in RCEW-II and 32.74% in RCEW-III”. Could the authors clarify what this percentage refers to? Is it the percentage of PHA per dry cell weight (w/w)?

Exactly, the percentage refers to intracellular PHA content, hence the percentage of PHA per dry cell matter (% PHA/CDM).

19) 3.4. Characterization of the PHA (Line 365) – Please, specify what HB is

HB stands for hydroxybutyrate, the acronym was explained in the text.

20) Please, use italic format for the name of bacteria. Some examples where this is not followed are in Line 334, 353, 361, 366 and 370. Check the manuscript and correct it.

Ok. All bacteria names are in italic.

21) 4. Discussion – Numbers in Figure 3 cannot be read. Please, provide a better image quality.

Figure 3, with clearer axis titles and color bars, was updated.

Suggestions

  • The title is too long and some keywords appear on it (microorganism and product). I suggest to reduce the title not using the keywords if possible. On the other hand, “Substrate optimization” appears among the keywords. Was it the sole parameter study?

The reviewer is correct, the substrate was the main parameter studied; hence it could be more appropriate to use it in the title instead of “optimization of biotechnological processes”

Maybe a better title and keywords could be:

Title:     Substrate optimization valorizing ricotta whey into PHB using Azohydromonas lata

Keywords:         dairy     waste   valorization;     polyhydroxybutyrate;  bioplastic (biodegradable polyester, maybe?); nutrient limitation strategy

Title and keywords were updated according to reviewer suggestions

Reviewer 2 Report

Comments and Suggestions for Authors

Major issues:

It has been well established that PHA-copolymers have desirable and superior characteristics that are suitable for biotechnological applications compared to PHB.
Several studies based on whey as a feed have successfully reported the production of PHA copolymers - P(3HB-co-3HV). Most of these studies have reportedly achieved cell biomass concentrations in the 16.8–24 g/L range with PHA yields between 12 and 14.7 g/L. On the other hand, the present study reports significantly lower biomass and PHB yields, which is a significant limitation.

Further, sterilization is an energy-intensive step. Many researchers have demonstrated the feasibility of using biowaste as feed for PHA production under non-sterile conditions to make the process economical, since feed contributes to 45% of the production cost. 

Additionally, the manuscript is unnecessarily long, and several experimental parameters carried out and presented seem to lack any relevance or scientific justification.

Author Response

Major issues:

It has been well established that PHA-copolymers have desirable and superior characteristics that are suitable for biotechnological applications compared to PHB.
Several studies based on whey as a feed have successfully reported the production of PHA copolymers - P(3HB-co-3HV). Most of these studies have reportedly achieved cell biomass concentrations in the 16.8–24 g/L range with PHA yields between 12 and 14.7 g/L. On the other hand, the present study reports significantly lower biomass and PHB yields, which is a significant limitation.

We agree that the production of copolymers such as P(3HB-co-3HV) is desirable due to their superior properties compared to PHB. However, Azohydromonas lata DSM 1123 requires the presence of specific precursors (e.g., propionic acid, valeric acid, or other volatile fatty acids) in the substrate to synthesize the 3-hydroxyvalerate (3HV). In this study, we deliberately avoided the supplementation of such precursors in order to pursue a process as simplified as possible in terms of pretreatments and operational complexity. The goal was to evaluate the PHB production potential of A. lata DSM 1123 under low-input conditions (reduced in both material costs and labour) by relying solely on minimal mineral supplementation.

This rationale, as well as the broader aim of developing a more scalable and cost-effective process, has been further clarified in the revised manuscript.

As the reviewer correctly notes, the PHA yields and biomass concentrations obtained in this study are lower than those reported in other works using whey substrates. However, so too are the associated costs of process supplementation, pretreatment, and operational management. While we do not yet have sufficient data to definitively conclude whether a simplified, low-yield process is more advantageous than a more controlled, high-yield one (a techno-economic analysis would be required) this study contributes to the existing literature by exploring a strategy that could be more easily implemented in non-dedicated facilities, such as dairy plants or wastewater treatment units.

Additional comments have been included in the revised version of the manuscript to better highlight the aims, novelty, limitations and implications of this work.

Further, sterilization is an energy-intensive step. Many researchers have demonstrated the feasibility of using biowaste as feed for PHA production under non-sterile conditions to make the process economical, since feed contributes to 45% of the production cost. 

Certainly, the evaluation of a less energy-intensive process is worth considering. However, beyond the issue of microbial contamination, in this case we also aimed to apply a treatment that could reduce the protein content of the substrate, which is known to hinder PHB accumulation by the microorganism. Currently, such protein removal can be achieved but typically requires costly treatments or advanced technological infrastructure (e.g., membrane filtration) (https://www.mdpi.com/2304-8158/9/10/1459) whereas the sterilization can be easily performed even in the dairy facilities since plate heat exchanger systems already exist there.

Nevertheless, in line with your suggestions, we have highlighted the limitations associated with thermal treatments in the revised manuscript and added new comments addressing the need to explore valid alternative strategies as part of future research directions.

Additionally, the manuscript is unnecessarily long, and several experimental parameters carried out and presented seem to lack any relevance or scientific justification.

Several paragraphs were shortened and each parameter modified within the One-Variable-at-a-Time approach was justified.

Reviewer 3 Report

Comments and Suggestions for Authors

The study optimized the process in PHA production from demonstrated feedstocks by DSM 1123, and found that the pre-fermentation treatments and protein/lactose/mineral supplementation in the media affect the PHA production. In general, the study is lack of novelty as the research on candidate substrates for one well-known model strain can not provide academic contribution towards known techniques.  In addition, the proposed feeding substrates do not significantly improve the titers and yields, so what's the necessary of the proposed technique?

In aspects of writing part of the manuscript. The main context including experimental methods, results, discussion and tables/figures are well prepared with clear data and logical structure. One minor comment is that the authors could adde one more comprehensive table which reviews and lists the main outcome of the strain DSM 1123, thus make it easier for the reader to judge best feedstocks the more economic methods for PHA production. 

Author Response

The study optimized the process in PHA production from demonstrated feedstocks by DSM 1123, and found that the pre-fermentation treatments and protein/lactose/mineral supplementation in the media affect the PHA production. In general, the study is lack of novelty as the research on candidate substrates for one well-known model strain can not provide academic contribution towards known techniques.  In addition, the proposed feeding substrates do not significantly improve the titers and yields, so what's the necessary of the proposed technique?

Ok. We acknowledge that Azohydromonas lata DSM 1123 is a well-known PHA-producing strain and that many studies have explored its behavior on various substrates. However, the novelty of this study lies not in the use of the microorganism itself, but in the development of a simplified and potentially scalable process using a poorly exploited dairy waste stream, ricotta cheese exhausted whey (RCEW), as a feedstock.

Unlike many previous studies that rely on pure substrates or extensively treated whey, this work evaluates PHB production from RCEW with minimal pretreatment, limited mineral supplementation, and without the addition of costly precursors or chemical inducers. The aim is to propose a low-cost and low-input process that could be realistically implemented in non-dedicated facilities such as dairy plants or wastewater treatment units.

Although the achieved yields are lower than those reported for more refined systems, the process simplification and the use of a waste-derived substrate represent a step toward practical applicability and circular economy.

Several new comments, included in the revised manuscript, now better emphasize this perspective and its potential relevance to industrial sustainability.

In aspects of writing part of the manuscript. The main context including experimental methods, results, discussion and tables/figures are well prepared with clear data and logical structure. One minor comment is that the authors could adde one more comprehensive table which reviews and lists the main outcome of the strain DSM 1123, thus make it easier for the reader to judge best feedstocks the more economic methods for PHA production. 

Tables 2 and 3 review and lists the main outcome of A. lata DSM 1123 performances during all tested conditions

Reviewer 4 Report

Comments and Suggestions for Authors

The study shows PHA could be augmented from cheese whey. This study is intriguing and falls well within the scope of the journal. The authors can improve the quality of their work by taking into these suggestions:

Abstract: Pasteurized RCEW or sterilized RCEW, use the correct terminology.

Introduction: Indeed, the study of biopolymers is a well-established technological route. The authors need to state what is new/novel about their study. How are they contributing to the overall scientific body of knowledge?

Results and discussion:

1. Page 9, line 353: A. lata DSM1123 should be in italics.

2. Which RSM tool was used for optimization?

3. Any validation/optimization results from the model?

4. The shortcomings of these results need to be clearly discussed. These could serve as a basis for further studies in this technological domain.

Author Response

The study shows PHA could be augmented from cheese whey. This study is intriguing and falls well within the scope of the journal.

The authors thank the reviewer for the comments and suggestion. A point-by-point revision was provided.

The authors can improve the quality of their work by taking into these suggestions:

Abstract: Pasteurized RCEW or sterilized RCEW, use the correct terminology.

Both pasteurized and sterilized RCEW were used with different results

Introduction: Indeed, the study of biopolymers is a well-established technological route. The authors need to state what is new/novel about their study. How are they contributing to the overall scientific body of knowledge?

Ok. We acknowledge that Azohydromonas lata DSM 1123 is a well-known PHA-producing strain and that many studies have explored its behavior on various substrates. However, the novelty of this study lies not in the use of the microorganism itself, but in the development of a simplified and potentially scalable process using a poorly exploited dairy waste stream, ricotta cheese exhausted whey (RCEW), as a feedstock.

Unlike many previous studies that rely on pure substrates or extensively treated whey, this work evaluates PHB production from RCEW with minimal pretreatment, limited mineral supplementation, and without the addition of costly precursors or chemical inducers. The aim is to propose a low-cost and low-input process that could be realistically implemented in non-dedicated facilities such as dairy plants or wastewater treatment units.

Although the achieved yields are lower than those reported for more refined systems, the process simplification and the use of a waste-derived substrate represent a step toward practical applicability and circular economy.

Several new comments included in the revised manuscript (especially in the discussion section) now better emphasize this perspective and the potential relevance to industrial sustainability.

Results and discussion:

  1. Page 9, line 353: A. lata DSM1123 should be in italics.

Ok. All bacteria names are in italic.

  1. Which RSM tool was used for optimization?

The modeling of cell growth and PHA accumulation as a function of protein content and mineral supplementation was performed using Statistica 12.5 (StatSoft Inc., Tulsa, OK, USA). This is reported in the statistical analysis paragraph.

  1. Any validation/optimization results from the model?

The model allowed the evaluation of the effects of linear, quadratic and interactive terms of the independent variable (protein content and mineral supplementation) on the dependent variables (cell growth and PHA accumulation). The results confirmed that moderate nitrogen limitation is beneficial for intracellular PHA accumulation and a progressive reduction of protein concentration, correlated with increased intracellular PHA content. This supports the well-established model in which nitrogen limitation combined with abundant carbon triggers PHA biosynthesis as a carbon and energy storage strategy. Still extreme nitrogen depletion along with low level of mineral supplementation impair both cell dry matter and PHA accumulation.

  1. The shortcomings of these results need to be clearly discussed. These could serve as a basis for further studies in this technological domain.

These aspects were discussed in lines 446-457

Reviewer 5 Report

Comments and Suggestions for Authors

This manuscript presents a comprehensive and methodologically rigorous investigation into the use of ricotta cheese-exhausted whey (RCEW) as a substrate for PHB production using Azohydromonas lata DSM1123. The study combines physicochemical characterization, process optimization, bioreactor validation and polymer characterization. The novelty lies in the valorization of a specific dairy by-product (RCEW) and in the process optimization via simple and scalable thermal and mineral interventions.

The manuscript is well-structured and supported by detailed experimental data. The figures and tables are informative and mostly well-prepared. However, the following issues must be addressed to improve clarity, reproducibility, and scientific rigor:

ABSTRACT

Abstract is dense and overly long; consider condensing into 200–250 words.

Clarify what is novel: the process optimization? The use of RCEW? The scalability?

Replace "ricotta cheese exhausted whey" with "RCEW" after first mention.

INTRODUCTION

The introduction provides a strong justification for PHB research but is excessively broad in parts (lines 37–66). Consider streamlining.

The rationale for choosing A. lata DSM1123 (e.g., specific productivity, suitability for complex substrates) should be stated more explicitly.

Line 50: change “the advancement of alternative materials” by “the development of alternative materials”.

Remove repeated citation URLs in refs [7–9].

MATERIALS & METHODS

Missing some key aspects:
– Provide the exact concentration of lactose hydrolysis products in pre-adaptation media.
– Clarify sterilization details (e.g., what pressure during 121 °C?).
– Bioreactor conditions: define pOâ‚‚ (if controlled), foam control, and feeding rate in fed-batch.
– State clearly if measurements were biological replicates (n = 3) or technical duplicates.

Check for consistent units (e.g., "g/L" vs. "% w/v").

Correct typos: "cell densitiy" (line 141)

RESULTS

Table 2 is very rich but overwhelming; suggest breaking into two subtables (e.g., flask vs bioreactor).
Statistical differences are claimed (e.g., p < 0.05), but no post-hoc test is mentioned. Specify test (Tukey, Duncan, etc.).

Figures are generally of good quality, but Figure 2 and 3 need clearer legends and axis labeling.

DISCUSSION

Good discussion on nitrogen limitation and PHB accumulation. However the authors should discuss the environmental/economic relevance of their thermal and mineral approaches (e.g., cost per liter).

FIGURES AND TABLES

Figure 3: Use clearer axis titles and color bars; the 3D surface plots are useful but hard to interpret without contextual explanations.
Figure 2: Units for DTG axis are missing. Add clearer label.

Table 1: consider summarizing chemical vs microbiological data in separate sections.

Author Response

This manuscript presents a comprehensive and methodologically rigorous investigation into the use of ricotta cheese-exhausted whey (RCEW) as a substrate for PHB production using Azohydromonas lata DSM1123. The study combines physicochemical characterization, process optimization, bioreactor validation and polymer characterization. The novelty lies in the valorization of a specific dairy by-product (RCEW) and in the process optimization via simple and scalable thermal and mineral interventions.

The manuscript is well-structured and supported by detailed experimental data. The figures and tables are informative and mostly well-prepared.

The authors thank the reviewer for the comments and suggestion. A point-by-point revision was provided.

However, the following issues must be addressed to improve clarity, reproducibility, and scientific rigor:

ABSTRACT

Abstract is dense and overly long; consider condensing into 200–250 words.

As per the journal guidelines, the abstract is 200 words long

Clarify what is novel: the process optimization? The use of RCEW? The scalability?

We acknowledge that Azohydromonas lata DSM 1123 is a well-known PHA-producing strain and that many studies have explored its behavior on various substrates. However, the novelty of this study lies not in the use of the microorganism itself, but in the development of a simplified and potentially scalable process using a poorly exploited dairy waste stream, ricotta cheese exhausted whey (RCEW), as substrate.

Unlike many previous studies that rely on pure substrates or extensively treated whey, this work evaluates PHB production from RCEW with minimal pretreatment, limited mineral supplementation, and without the addition of costly precursors or chemical inducers. The aim is to propose a low-cost and low-input process that could be realistically implemented in non-dedicated facilities such as dairy plants or wastewater treatment units.

Although the achieved yields are lower than those reported for more refined systems, the process simplification and the use of a waste-derived substrate represent a step toward practical applicability and circular economy.

Several new comments, included in the revised manuscript, now better emphasize this perspective and its potential relevance to industrial sustainability.

Replace "ricotta cheese exhausted whey" with "RCEW" after first mention.

RCEW was used throughout the text

INTRODUCTION

The introduction provides a strong justification for PHB research but is excessively broad in parts (lines 37–66). Consider streamlining.

As suggested, the paragraph was shortened.

The rationale for choosing A. lata DSM1123 (e.g., specific productivity, suitability for complex substrates) should be stated more explicitly.

Azohydromonas lata DSM 1123 was selected for this study due to its well-documented ability to produce high levels of PHB under relatively simple cultivation conditions. Compared to other PHA-producing microorganisms, A. lata does not require a strictly defined two-stage process for biomass accumulation followed by polymer synthesis. Instead, PHB accumulation in this strain tends to be growth-associated and occurs proportionally to the growth rate under appropriate nutritional conditions. This metabolic behavior enables the design of simplified processes without the need for tightly controlled nutrient limitation strategies. In addition, A. lata is known for its fast growth rate, broad substrate utilization, including lactose-rich dairy waste streams, and tolerance to suboptimal or non-sterile environments, which further supports its suitability for low-cost, scalable applications using minimally pretreated feedstocks such as ricotta cheese exhausted whey (RCEW).

New comments with proper references have been added to the revised text.

Line 50: change “the advancement of alternative materials” by “the development of alternative materials”.

Ok.

Remove repeated citation URLs in refs [7–9].

 As per the journal guidelines URLs were added to the references

MATERIALS & METHODS

Missing some key aspects:
– Provide the exact concentration of lactose hydrolysis products in pre-adaptation media.
– Clarify sterilization details (e.g., what pressure during 121 °C?).
– Bioreactor conditions: define pOâ‚‚ (if controlled), foam control, and feeding rate in fed-batch.
– State clearly if measurements were biological replicates (n = 3) or technical duplicates.

  • a new comment regarding the hydrolysis degree of lactose has been added in paragraph 2.3.
  • details about the pressure conditions of the sterilization have been added (2.3.3)
  • the missing details for the bioreactor conditions have been added to the paragraph 2.5
  • In paragraph 2.7, it was reported that all data of biochemical analyses were obtained at least three biological replicates, and each replicate was analyzed twice

Check for consistent units (e.g., "g/L" vs. "% w/v").

Units consistency throughout the text was checked.

Correct typos: "cell densitiy" (line 141)

Done. 

RESULTS

Table 2 is very rich but overwhelming; suggest breaking into two subtables (e.g., flask vs bioreactor).

As suggested table 2 was split.
Statistical differences are claimed (e.g., p < 0.05), but no post-hoc test is mentioned. Specify test (Tukey, Duncan, etc.).

Pair comparison of treatment means was performed by applying Tukey’s procedure at p < 0.05, this was specified in paragraph 2.7.

Figures are generally of good quality, but Figure 2 and 3 need clearer legends and axis labeling.

Figures were updated with clearer axis labeling and legends detailed

DISCUSSION

Good discussion on nitrogen limitation and PHB accumulation. However the authors should discuss the environmental/economic relevance of their thermal and mineral approaches (e.g., cost per liter).

Clearly, this study did not include a detailed cost-benefit or techno-economic analysis at this stage. However, the technical setup was deliberately designed based on principles of process simplification. For instance, thermal treatments were selected for their dual function: reducing nitrogen content and microbial contamination, thereby avoiding the need for more complex and costly pretreatment steps. Similarly, mineral supplementation was minimized to essential components only.

These strategies aim to reduce overall operational and labour costs, making the process more accessible and potentially scalable in non-specialized settings. The rationale behind these choices has been clarified and reinforced in both the methodological description and the discussion section of the revised manuscript.

FIGURES AND TABLES

Figure 3: Use clearer axis titles and color bars; the 3D surface plots are useful but hard to interpret without contextual explanations.

3D surface plots were discussed in lines 446-457 and the figure, with clearer axis titles and color bars, was updated.
Figure 2: Units for DTG axis are missing. Add clearer label.

Unit for the DTG curve (d(wt)/dT) was reported in figure 2 and the legend was clarified.

Table 1: consider summarizing chemical vs microbiological data in separate sections.

The reviewer’s suggestion is valid, the two sections could be separated, however, the table is pretty slim itself, probably, keeping it together would help the reader to have an overall idea of the chemical and microbiological composition of the RCEW.

Round 2

Reviewer 2 Report

Comments and Suggestions for Authors

The justifications provided do not contribute meaningfully to advancing the existing scientific knowledge in the published literature.

The present work and findings qualify at best for an M.Sc. dissertation.

Therefore, the study should focus on PHA copolymers, emphasizing whether they offer enhanced yields and improved material (physico-chemical) properties compared to those already documented.

Reviewer 3 Report

Comments and Suggestions for Authors

The application of a new feedstock for a well-know PHA production strain could not achieve "novelty and originaity" for a research paper unless promising improments are shown. In Author response to report ,  the authors debates that the proposed method is simplifier, lower cost, scalable in industrial production comparing with conventional process in PHA production. However, the curent tables (2/3) only list the own data and did not PRESENT SUCH COMPARISON and justify such improvement comparing with conventional process (as suggested in version one comment).